# The Impact of Religion and Social Support on Self-Reported Happiness in Latin American Immigrants in Spain

Angélica M. Formoso-Suárez [1], Jesús Saiz [1,*] , Deepak Chopra [2] and Paul J. Mills [2]

1    Department of Social, Work and Differential Psychology, Complutense University of Madrid,
     28040 Madrid, Spain; aformoso@ucm.es
2    Department of Family Medicine and Public Health, University of California San Diego, La Jolla, CA 92093,
     USA; nonlocal@deepakchopra.com (D.C.); pmills@health.ucsd.edu (P.J.M.)
*    Correspondence: jesus.saiz@psi.ucm.es

**Abstract:** The Latin American immigrant population in Spain is growing year after year. The aim of this study is to describe how religious coping mechanisms, religiosity, acculturation and social support act in the prediction of happiness within the group of Latin American immigrants with Christian beliefs in Spain. Previous studies indicate that religious practice can be a factor that favors the perception of well-being and happiness in people. It is considered that the existence of close sources of support act as a key resource in facing stressful situations. A questionnaire consisting of 36 items was administered to a sample made up of 206 subjects of different Latin American nationalities who were going through a migration process. It was observed that religiosity, social support, religious coping mechanisms and gender were related to the perception of happiness. The results underline the importance of considering religiosity and social support together while intervening to improve the perception of happiness of Latin American immigrants in Spain.

**Keywords:** religiosity; happiness; religious coping mechanisms; social support; immigration





## 1. Introduction

Spain is a country that, due to its historical context, has received large numbers of Latin American immigrants over the years. Likewise, it has been a geographical point from which many natives have gone abroad in search of an improvement in their living conditions. The Spanish migration statistics produced by the INE (2021) regarding the entry of people of Latin American origin into the country indicate that over the years the number of people who reside in Spain who originate from the American continent is increasing (Figure 1).

Some statistics also show that the countries that generate a greater flow of people to Spain from South America have been Colombia and Venezuela (INE 2021) (Figure 2). Other analyses conducted by the Pew Research Center (2018) indicate that Latin America and the Caribbean constitute a population that for the most part would be considered Christian with high levels of affiliation to their religion. Within these countries, the ones that stand out mainly of people affiliated to their religious beliefs are Honduras (90% of affiliation) and Nicaragua (93% of affiliation).

Having a greater understanding of the religious beliefs of these immigrants to Spain can provide clues for new research and potential interventions. Specifically, it would be very helpful to understand in greater depth the coping mechanisms of a religious nature that are used to deal with the migratory process, as well as the happiness and overall well-being the individuals derive from their religion.

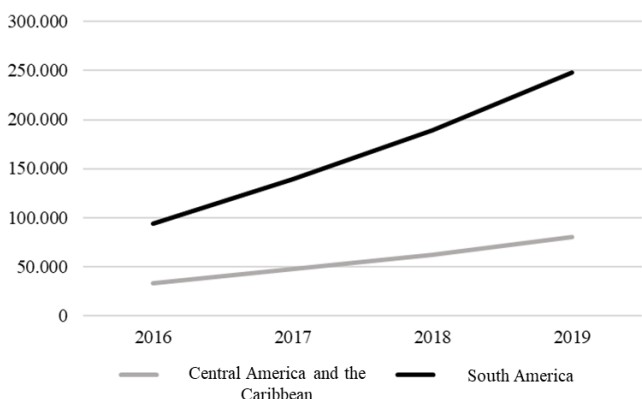

**Figure 1.** Immigration of Latin American population in Spain over the years. The data are provided by the INE (2021).

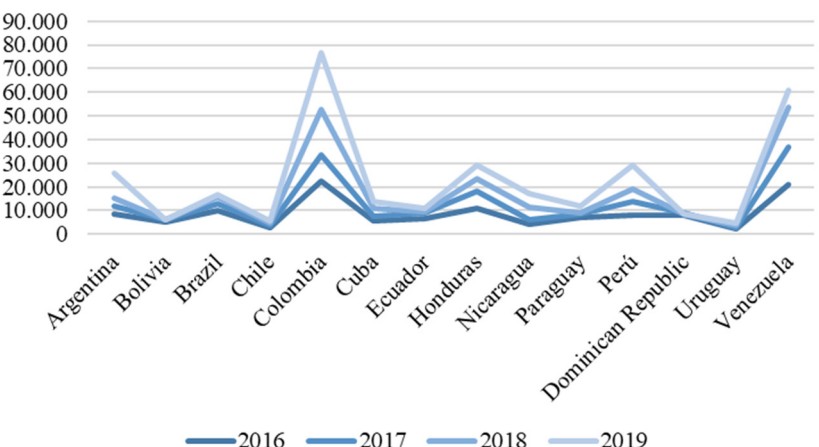

**Figure 2.** Number of Latin American immigrants in Spain by country of origin. The data is provided by the INE (2021).

## 1.1. Religion and Religious Coping Mechanisms

Numerous studies demonstrate that religion offers individuals an increased capacity to cope with loss. Pargament et al. (2000), for example, compiled a series of purposes and benefits of practicing a religion, and reported factors such as life meaning, control over situations, comfort, intimacy and vital transformation. Geertz (1966) argued that religion endows our lives with meaning, showing that carrying out religious practices allows us to discover and give meaning to our actions and our own life course, enhances our ability to deal with unjust situations and gives us the opportunity to generate deeper analyses about the meaning of living. Meanwhile, Fromm (1950) stated that religion serves as a controlling agent that provides a locus of external causality, explaining itself through the existence of a divine entity that decides for us. In line with the above, in stressful situations, religion can act as a coping mechanism for the person, generating attributions to an external agent that allows the burden of the circumstance for the individual to be reduced. Regarding consolation, and related to the previous function, Johnson (1959) pointed out that religion allows us to connect with a superior force, which is usually key within one's own belief and practice of faith. In addition, religion has an inherent cooperative character that allows individuals to behave in an altruistic fashion and collaborate with others. It also provides a sense of community, reinforcing group identity and closeness to God (Buber 1970).

Regarding other functions of religion, Pargament (1997) points out that it fulfills a fundamental role in facing life changes and in facing challenges. Religious practice and the values that underlie such practices allow the individual to engage and transform their life. This suggests that people rely on religion to modify their own behaviors and seek

meaning in even the most stressful situations. Believers reconstruct the circumstances, endowing them with meaning associated with their own beliefs. This allows them to reinterpret the circumstances in different ways. Pargament emphasizes that religion acts as an active mechanism that contributes to overcoming existential dilemmas, providing solutions and as a basis for solving problems. Authors such as San Román et al. (2019) have found that people who share certain behaviors of a religious nature have high resolution capacity and a better quality of life due to the resilience developed thanks to their religious beliefs. In other words, religion acts as a factor that provides coping mechanisms in adverse situations.

It has been found that religiosity provides protective factors to individuals, notably improving both their health and the ways of relating to others (Saiz et al. 2020). Conversely, religion has been found to enable individuals to cope well with illness and disability (Salgado 2014). Religion seems to provide a greater subjective perception of well-being and happiness (Salgado 2014) which is related to its functions in terms of generating a sense of purpose and the emergence of values such as kindness and compassion (Saiz et al. 2021a). Likewise, religion seems to have an influence on the individual's self-perception, improving their self-esteem and triggering a series of prosocial actions towards their environment. Religious practice favors healthy lifestyles, such as reducing the consumption of toxic substances including alcohol and drugs and promoting behaviors that keep people away from other risk factors (De Diego and Guerrero 2018). In addition, religion has positive effects on hope and resilience, which are relevant to the development of an individual's coping capacity (De Diego and Guerrero 2018).

According to Yoffe (2012), having faith in a higher being and the trust attributed to one's own religious practice seem to be factors that can affect the perception and management of stress in the face of grief. In the same way, praying to God generates tranquility, whether this is a request or an act of gratefulness to the higher entity. Both individual and group religious practices give believers confidence in their God and their religion. This allows adverse situations to be dealt with more effectively despite their difficulty. Yoffe (2012) points out that religious practice can generate positive effects on four different levels: psychological states, emotions, cognitions and behaviors. As Wortmann (2013) defined, "religious coping is religiously framed cognitive, emotional, or behavioral responses to stress, encompassing multiple methods and purposes as well as positive and negative dimensions".

However, religion can also be lived in different ways. In studies such as the one conducted by Cetrez (2011), it was found that there are differences between men and women when it comes to religious practice. Women display more religious behaviors than men, one of the main ones being fasting. Pargament (1997) indicates that people tend to be situated on a continuum of religiosity made up of different behavioral levels of religious practice. There are people who feel more connected to God through the exercise of prayer and people who are more connected on those occasions in which they share activities related to spirituality within their own religious community. Pargament (1997) proposes that there is no single definition of religion, since each individual and community experiences religion in different ways. Religion has practical, symbolic and experiential components defined by the practitioners themselves and the ecclesiastical institution, allowing for greater cohesion and connection with spirituality. Although religion is often associated with the more institutional level (public dimension) and religiosity with personal experience (private dimension), here we will identify religiosity with religion and follow Koenig et al. (2012), who state that religion is an organized system of beliefs, practices, rituals and symbols designed to facilitate closeness to the sacred or transcendent, where the transcendent varies according to Western religious traditions (God, Allah, a Higher Power) and Eastern traditions (Buddha, Brahman, the ultimate truth). In this particular study, we will focus on Christianity since it is the most practiced religious among the Latin American population (Pew Research Center 2018).

### 1.2. Social Support and Religious Community

Some authors understand social support as interpersonal exchanges that involve help, affection and affirmation (Khan and Antonucci 1980). This set of exchanges between individuals that operates on social networks can be presented through resources, actions and information. This is how four different supportive domains can be distinguished: material, instrumental, emotional and cognitive (Guzmán et al. 2003). Material support involves the exchange of monetary resources or other goods, such as clothing and food. Instrumental support is one that starts from helping another person in some type of task. Emotional support is that which is born from the shared bond between individuals and which in turn leads to trust, understanding and affection. Finally, when talking about cognitive support, emphasis is placed on the transmission of information from one to another.

When people are part of a group, relationships and social interactions are generated with its members. In this specific case, being part of religious groups may constitute a source of social support. Through the relationships established in the groups, important information resources can be obtained, as well as allowing greater integration in the community with positive implications for both individuals and social well-being (Cohen et al. 2000). This social integration is made up of three levels, explaining the links between people and their environment. Lin (1986) points out that these levels are the community, social networks and intimate and trusting relationships. The community denotes the integration of the person in an environment, which provides a feeling of belonging to a more complex social structure that at the same time provides a sense of belonging. Moreover, social networks emerge. These become associated with family, work and friends. These relationships entail a reasonable frequency and level of commitment by the individuals involved in them. Finally, intimate networks generate commitment and search for the well-being of the other, which again encourages altruism among members of the same group (Lin 1986).

Regarding depression, it has been shown that migrants show a slight tendency to have higher depression scores than non-migrants. However, it has been found that social support networks during periods of individual adjustment are essential to oppose the development of depressive symptoms (García et al. 2001). In these settings, social support can thus act as a factor that protects individuals, contributing to facing challenges and solving somehow emotional, informational and material needs (García et al. 2001). Some studies, such as the one conducted by Palomar and Cienfuegos (2007), indicate that religious communities and God's belief are very important sources of social support. In many cases, religious communities constitute a point of support where individuals seek help in situations that cause them sadness. It has been shown that those people who rely more on church and religion have an external locus of control, emphasizing dependence on their God for success. It has also been found that people perceive higher levels of social support when going to the religious community while considering that their well-being can come from being pleasant to others. It seems that, although family is the main source of support, people perceive that their church provides greater support compared to friends and neighbors that are part of their communities (Palomar and Cienfuegos 2007).

In line with this, Connor and Koenig (2013) found that, in contexts with a thriving religious field (such as the United States), religious attendance tends to be positively related to occupational attainment, especially for the second-generation immigrants. In addition, the effect of religious affiliations for social ties was also observed in youth by Simsek et al. (2020), who noticed that social ties are more likely between classmates with similar religious affiliations than classmates with different religious affiliations.

### 1.3. Happiness

Happiness and religions have been the subject of multiple studies, with findings consistently showing a positive influence of religious beliefs on happiness. Authors such as Argyle et al. (1995) propose happiness as a multidimensional construct where positive

emotions, satisfaction and the absence of negative emotions act determining people's level of happiness. According to Veenhoven (1995), happiness corresponds to a favorable judgment made on the quality of life. If we unite both definitions, we could say that happiness is a term that involves aspects of both emotional and perceptual origin; that in turn includes the individual's own assessment of their quality of life. Happiness has attitudinal components where people assess their lives, cognitively comparing the state in which they find themselves with the desired one. Therefore, happiness is an indicator resulting from the comparison between real life and the ideal life. In this way, we could point out that the smaller the difference between the life we want and the life we have, the greater our perception of happiness (Abdel-Khalek 2006).

Some studies, such as that of Francis et al. (2012), point out that the relationship between religion and happiness is dependent on the geographical area where the person is located and their individual characteristics. For example, in countries such as Estonia, the results show the lack of relationship between religion and happiness. In other words, practicing a religion is not relevant when determining a person's well-being.

Nevertheless, after an exhaustive bibliographic review, Myers (2008) argues that people who practice some type of religion and that are involved in activities related to their beliefs tend to show higher indicators of satisfaction with life and happiness as compared to people who do not carry out any kind of religious practice. In fact, in a survey directed by the National Opinion Research Center in 2006 (as cited in Myers 2008), no differences were observed between Protestant, Catholic and Jewish people when it came to indicating their levels of happiness with life. It was also found that, among Americans, those who observe some type of religious practice that made them feel close to their God showed the highest levels of happiness as compared to those without such practices. Consistent with that observation, Abdel-Khalek (2006), in a population sample from Kuwait, concluded that although the main predictor of happiness was mental health, religion was among the elements that led to individuals feeling more or less happy.

This was also demonstrated in a longitudinal study carried out in Germany. Headey et al. (2008) suggested that people who have religious beliefs and who throughout their lives carry out more altruistic and voluntary activities reach higher subjective levels of happiness. These authors also indicated that people who get involved in the activities of their religious community can increase the level of perception of satisfaction regarding their state of well-being.

Considering the continually growing immigration of Latin American people in Spain, it becomes necessary to study how religion affects their happiness and perceived well-being to better understand how they adapt to their new context. The primary aim of this study is to describe how religious coping mechanisms, religiosity and social support act in the prediction of happiness within a group of Christian Latin American immigrants in Spain. In addition, we will also study, according with previous findings in other countries, as a general hypothesis, the existence of religious coping mechanisms and their positive impact on migratory processes, as well as the benefit of religious practice in the perception of happiness of individuals are expected.

## 2. Method

### 2.1. Design

This study follows a correlational research design and employs a convenience sampling procedure. The participants were contacted by email, WhatsApp conversations and other social networks such as Facebook, Twitter and Instagram. Once they agreed to participate, they were sent a link that directed them to a questionnaire on the Google Forms platform. Before starting to respond, it was mandatory to give consent for the handling of the data. Therefore, all the subjects who completed the questionnaire gave their approval before participating and were informed about the aim of the study.

The questionnaire was administered for 15 days to the target population (from 5 April to 20 April 2021). Once the form was completed, the responses were compiled into an

Excel file and the form was cleaned, deleting the cases that were incomplete or that didn't correspond to the questions given to the participants. The answers given by participants who didn't fit into the sample were also discarded. After gathering enough information, the database was cleaned using the Excel program. After cleaning the data, the pertinent analyses for the study were conducted. Once the analyses were completed, those study participants who requested information about the study were contacted by email to receive the results.

*2.2. Sample*

Participants were selected through direct contact with religious groups from Christian parishes from Spain, most of them being from the region of Madrid. There were 206 subjects selected for convenience, based on the following criteria: country of origin (Latin American origin), religious belief (Christian-type beliefs), current residence in Spain and majority age (+18). The sample was made up of 131 women (63.6%) and 75 men (36.4%) whose ages ranged from 18 to 74 years (M = 35.86; SD = 14.07). The mean age of the final sample was 36 years. Regarding nationality, the individuals who participated in the study came from different Latin American countries such as Argentina, Bolivia, Brazil, Chile, Colombia, Ecuador, Guatemala, Honduras, Mexico, Nicaragua, Paraguay, Peru, the Dominican Republic and Venezuela, the majority being from Venezuela (Figure 3). Regarding the time living in Spain, most of them had been residing in the country for at least one year, while all had been living here for at least six months and some had been here for 28 years, which was the maximum (M = 6.99; SD = 6.67). All participants had internet access to answer the questionnaire.

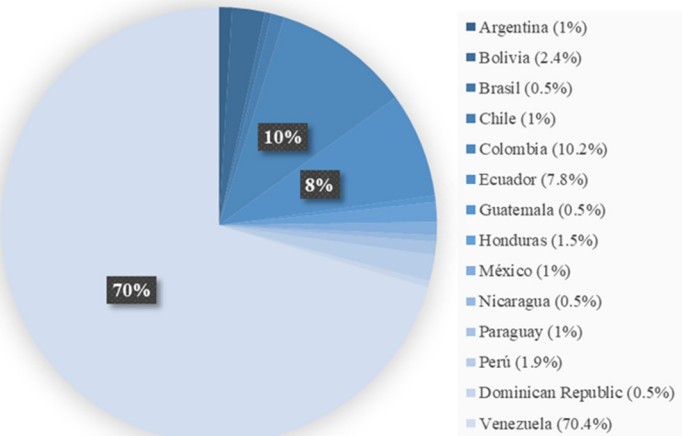

**Figure 3.** Country of origin.

*2.3. Variables and Instruments*

2.3.1. Sociodemographic Variables

Participants were asked about their gender and age. Gender was coded as 1 = Female and 2 = Male. The age variable was included in the analyses as a continuous. In addition, for descriptive motives, it was recalculated into another variable labeled Age Group, which divides the ages of our participants into four ranges: Group 1 (from 18 to 31), Group 2 (from 32 to 45), Group 3 (from 46 to 59) and Group 4 (from 60 to 74). These categories were established after consulting the parameters used by the Center for Sociological Research (Center for Sociological Research 2021). The participants were also asked about the religion they believe in. In this case, only subjects whose beliefs were based on Christianity (Catholic, Protestant or others) were selected. In addition, they were asked about their belonging to religious groups. Hereby, it was possible to distinguish between people who held beliefs and belonged to religious groups and people who declared to have religious beliefs but did not belong to any religious group or parish.

### 2.3.2. Social Support

To assess perceived social support, 12 items of the Social Support Scale (EAS) by Palomar et al. (2013) were used. This scale aims to obtain indicators that reflect the degree of support that people perceive from their closest social circle. Composed of 21 items, the Social Support Scale evaluates three factors (Social Support, Family Support and Support from Friends), explaining 66.24% of the total variance. The items are divided as follows for each factor: there are 12 for social support ($\alpha$ = 0.948), 6 items for family support ($\alpha$ = 0.897) and 3 items aimed to measure support from friends ($\alpha$ = 0.714) (Palomar et al. 2013). On this occasion, only the items corresponding to factor 1 were used. Factor 1 has been selected to prepare the final instrument since it collects generic information on the quality and existence of social ties without the need to label the relationship that each individual shares with others in their social environment (family, friends, partner, community, etc.). An example of an item for this factor would be the following: "I have at least one person who has been with me in difficult times". The responses are placed on a Likert-type scale that ranges from 1 indicating that they "Strongly disagree" with the statement made and "4" indicating that they "Strongly agree". This instrument has been chosen because of its language, brevity and reliability in measuring. When applied to our sample, the scale showed acceptable levels of reliability ($\alpha$ = 0.896).

### 2.3.3. Coping Mechanisms of Religious Origin

To quantify the coping mechanisms of religious origin, the questionnaire of religious coping mechanisms (R-COPE) validated into Spanish by Martinez and Sousa (2011) was used. This instrument has an acceptable validity related to the variable measured ($\alpha$ = 0.86). For the Spanish validation, these levels of reliability remained high ($\alpha$ = 0.864). The scale is made up of 12 items, divided into two factors: Positive Religious Coping (6 items) and Negative Religious Coping (6 items). Factor 1 explains 33.3% of the variance, while Factor 2 explains 29.1% of it. The following is an example of an item related to Factor 1: "I looked for a stronger connection with God". Regarding Factor 2, an example of an item is the one presented below: "I wondered if God had abandoned me". The answers are presented in a Likert-type scale that oscillates between 0 "Never" and 3 "A lot", showing the degree of agreement with the statement offered. In order to facilitate the interpretation of the results, the items corresponding to Factor 2 were inverted, with which higher scores refer to the absence of negative strategies. As in the original scale produced by Pargament (1997), and for the Spanish validation used here, both factors showed high test–retest reliability, yielding indicators of r = 0.78 and r = 0.72 respectively (Martinez and Sousa 2011). When applied to our sample, the scale showed acceptable levels of reliability ($\alpha$ = 0.865).

### 2.3.4. Religiosity

The Duke University Religion Index (DUREL) scale made by Koenig and Büssing (2010) was used to quantify religious practices. This instrument is made up of a total of five items with different purposes among them. This scale seeks to measure organizational religious activity (ORA), specifically to what extent the person attends public and group activities related to their religion. Likewise, the scale measures the frequency of non-organizational religious activities (NORA), which refers to all behaviors and practices carried out at the private/individual level. This scale also measures intrinsic religiosity (IR), which refers to the degree of individual commitment to religion. The scale has two items whose response format is a Likert-type scale that is delimited between 1 "Never" and 6 "More than once a week". The following is an example of an item found in the scale: "How often do you go to church or other religious gatherings?" On the other hand, the scale contains three remaining items whose Likert scale response format ranges from 1 "Definitely not" to 5 "Definitely true of me". An example of these is the following: "In my life, do I experience the presence of the Divine?" The scores obtained on this scale can be interpreted between 5 "Null Religiosity" and 27 "Intense Religiosity". The higher the scores, the higher the levels of religiosity.

Regarding its psychometric properties, the DUREL scale has been shown to have a test–retest reliability of r = 0.91 and a convergence validity with other scales of religiosity that ranges between r = 0.71 and r = 0.86 (Reyes-Estrada et al. 2014). The scale not only indicates the type of religiosity of the sample, but also the frequency of execution of behaviors related to it. This instrument has been translated into Spanish and validated with samples made up of Latin American people (Mexico) (α = 0.91) (Gómez-Alcalá et al. 2021). For our sample, the scale showed acceptable levels of reliability (α = 0.88).

### 2.3.5. Acculturation

To measure the acculturation level in the sample, the acculturation questionnaire validated into Spanish by Basabe et al. (2004) was used. It is composed of four items relating to both the country of origin and the host country. The different types of acculturation strategies are evaluated with this scale: biculturalism/integration, assimilation, separation and marginalization. An example of a question used in this questionnaire is the following one: "Do you think it is important to maintain the customs and traditions of your country of origin?" The response format oscillates on a Likert-type scale that goes from 0 "Not at all" to 10 "Very". Higher scores in the scale mean better adaptation to the host country without losing the traditions of the country of origin. This scale was used as it was a previously validated instrument in the Spanish language, presenting acceptable reliability (α = 0.83) and allowing information on the countries of origin and host countries to be obtained. However, when this instrument was applied in the sample studied here, the reliability levels were lower than expected (α = 0.68). With this in mind, it is necessary to analyze the results with certain levels of caution.

### 2.3.6. Subjective Happiness

To explore the subjective perception of happiness, the four items from the Subjective Happiness Scale of Lyubomirsky and Lepper (1999) validated in Spanish by Vera et al. (2011) were used. This scale generates a global measure of happiness. It also shows a high level of internal consistency (α = 0.86) and a convergent validity of r = 0.62 when compared with other happiness scales. An example of an item on the scale is the following one: "Some people are very happy in general: they enjoy life regardless of what happens to them. Do you consider yourself a person like the one just mentioned?" The response format of this scale is Likert-type, with a range of scores located between 1 and 7 with descriptions that vary depending on each item. This scale was used for its clarity and its psychometric properties.

### 2.4. Data Analysis

An initial descriptive analysis of the collected sociodemographic data was carried out. Following this, a correlation matrix was used among all the variables of the study to explore their relationships. Next, a linear regression test was performed to examine the influence of sociodemographic and independent variables on happiness as a dependent variable. A blocking procedure was followed. First, the influence of sociodemographic variables (gender, age and time living in Spain). Secondly, the influence of social variables was considered (acculturation and social support). Finally, the religious variables were introduced (belonging to a religious group, religious beliefs, religiosity and religious coping mechanisms). In addition, the VIF statistic was used as a measure of collinearity, considering values greater than 10 as problematic (Kleinbaum et al. 2014). All the analyses were performed with SPSS V.25.

## 3. Results

### 3.1. Sociodemographic Data and Characteristics of the Sample

Of the total sample, 144 (69.9%) considered themselves Catholic Christians, 37 (18%) Protestant Christians and 25 (12.1%) indicated that they did not belong to any of these categories (Other). Regarding the group membership of the sample, it was observed that

156 people (75.7%) did not belong to any religious group or parish. Of the 50 (24.3%) participants who answered "Yes" to this item, 68% indicated that they belonged to groups with Protestant Christian beliefs, while 32% indicated that their religious group was based on Catholic Christian beliefs (Table 1).

**Table 1.** Sociodemographic characteristics of the sample.

| Sociodemographic Characteristics | | *N* | % | % Accumulated |
|---|---|---|---|---|
| Sex | Feminine | 131 | 63.6 | 63.6 |
| | Masculine | 75 | 36.4 | 100 |
| Age category * | From 18 to 31 | 105 | 51 | 51 |
| | From 32 to 45 | 45 | 21.8 | 72.8 |
| | From 46 to 59 | 40 | 19.4 | 92.2 |
| | From 60 to 74 | 16 | 7.8 | 100 |
| Religious beliefs | Catholic Christian | 144 | 69.9 | 69.9 |
| | Protestant Christian | 37 | 18 | 87.9 |
| | Other | 25 | 12.1 | 100 |
| Group membership | No | 156 | 75.7 | 75.7 |
| | Yes | 50 | 24.3 | 100 |
| Membership group | None | 157 | 76.2 | 76.2 |
| | Catholic group | 15 | 7.3 | 83.5 |
| | Protestant group | 34 | 16.5 | 100 |

Note. N = 206. * Participants had an average age of 35.8 years (SD = 14.07).

The scores from the descriptive analyses indicate that most of the psychological variables showed normative parameters. The scores from religious coping mechanisms are higher for positive than for negative religious coping mechanisms (see Table 2).

**Table 2.** Psychological variables in the sample.

| Psychological Variables | Mean (SD) |
|---|---|
| Social support | 3.68 (0.46) |
| Acculturation | 7.89 (1.58) |
| Happiness | 5.43 (1.09) |
| Religiosity | 15.80 (6.82) |
| Positive religious coping mechanisms | 1.80 (1.13) |
| Negative religious coping mechanism | 0.71 (0.88) |

*3.2. Correlations between Independent Variables and Happiness*

It was found that the variables most closely related to each other are religiosity and positive religious coping mechanisms ($r = 0.789$, $p < 0.001$), and happiness with religiosity ($r = 0.392$, $p < 0.001$), social support ($r = 0.329$, $p < 0.001$), positive religious coping mechanisms ($r = 0.244$, $p < 0.001$) and negative religious coping mechanisms ($r = -0.206$, $p < 0.01$) (see Table 3).

**Table 3.** Pearson's correlations between psychological variables studied.

| | 1 | 2 | 3 | 4 | 5 |
|---|---|---|---|---|---|
| 1. Happiness | 1 | | | | |
| 2. Social support | 0.329 *** | | | | |
| 3. Positive religious coping | 0.244 *** | −0.027 | | | |
| 4. Negative religious coping | −0.206 ** | −0.233 ** | 0.169 * | | |
| 5. Religiosity | 0.392 *** | 0.009 | 0.789 *** | 0.034 | |
| 6. Acculturation | 0.095 | 0.118 | 0.126 | 0.031 | 0.039 |
| N | 206 | 206 | 206 | 206 | 206 |

Note. * $p < 0.05$; ** $p < 0.01$; *** $p < 0.001$.

*3.3. Regression Models for Happiness*

Different models were applied to understand the influence of sociodemographic and independent variables on happiness (Table 4). When testing the sociodemographic variables, age was the only significant variable [$R^2_{ADJ}$ = 0.014].

**Table 4.** Linear regression models for happiness.

| Models | Predictors | $R^2_{ADJ}$ | B | SE | *p* | 95% CI | | FIV |
|---|---|---|---|---|---|---|---|---|
| | | | | | | LL | UL | |
| 1 | Age | 0.014 | 0.011 | 0.005 | 0.048 | 0.00 | 0.21 | 1.00 |
| 2 | Social Support | 0.104 | 0.787 | 0.154 | 0.000 | 0.484 | 1.09 | 1.00 |
| | Age | 0.122 | 0.012 | 0.005 | 0.023 | 0.002 | 0.022 | 1.00 |
| 3 | Religiosity | 0.149 | 0.067 | 0.010 | 0.000 | 0.048 | 0.086 | 1.03 |
| | Social Support | 0.252 | 0.706 | 0.144 | 0.000 | 0.423 | 0.989 | 1.06 |
| | Negative Religious Coping | 0.271 | −0.164 | 0.076 | 0.031 | −0.313 | −0.015 | 1.08 |
| | Gender | 0.284 | 0.295 | 0.137 | 0.033 | 0.023 | 0.566 | 1.06 |

The second model tested sociodemographic variables, plus acculturation and social support. The results indicated that social support and age significantly predicted happiness ($R^2_{ADJ}$ = 0.122).

Finally, the third model, including also the variables belonging to a religious group, religious beliefs, religiosity and religious coping mechanisms, showed that religiosity, social support, negative religious coping and gender increased the percentage of explained variance to 28.4%.

## 4. Discussion

Pargament et al. (2000) already indicated that religion and its practice can bring different benefits to practitioners. Fromm (1950) pointed out the importance of religion as a reducing agent of pressure on an individual, focusing on God as an external locus of control. Other authors such as San Román et al. (2019) propose that religion allows altruistic activities to be carried out that involve the perception of quality of life and the development of resilience. In turn, religious communities seem to have an impact on the social support that the individual receives. This support can act as an agent that shapes the individual's ability to cope with difficult situations, allowing challenges to be overcome and certain needs to be fulfilled (García et al. 2001).

The aim of the present paper was to study how the variables of religious coping mechanisms, religiosity, acculturation and social support act in the prediction of happiness in Christian immigrants in Spain. When describing the study sample, it was found that most of the people from Latin American countries were not part of religious groups in the host country. However, despite not being involved in organizational activities of this type, Latin American immigrants defined themselves mostly as Christians. Regarding the variables of psychological origin, it was seen that people showed high scores for the variable coping mechanisms of religious origin, although they didn't carry out religious practices in community with others. This is closely related to what Yoffe (2012) raised about belonging to religious groups. The author pointed out that the way the practices are carried out is not relevant, but maintaining the belief in God is beneficial to face situations of uncertainty is highly relevant.

When analyzing the results obtained by the Pearson correlations test, it was found that the variables most related to happiness were religiosity, social support and positive religious coping mechanisms. These results underline the importance of religion and social support for the perception of happiness in the Latin American immigrant population in Spain. In addition, we found a negative correlation between negative religious coping mechanisms and happiness and social support. It can be inferred from this result that using

negative coping strategies risks people´s perception of happiness and their own social support systems. In this study we did not study the effect of using religious practices to hide feelings (spiritual bypassing), but this particular finding might be related to it (Motiño et al. 2021). Finally, it was found that positive religious coping mechanisms are greater in people whose religiosity is higher. In this way, the results suggest that those people who are most connected with their belief in God are the ones who rely the most on it.

Regarding the regression analyses, it was observed that variables such as religiosity, social support, negative religious coping mechanisms and gender could predict higher levels of happiness. In line with the results, religiosity is shown to be one of the most influential factors when it comes to perceiving happiness. It is all consistent with authors such as Salgado (2014), who have stated that religion and its practices provide purpose to people's lives, generating subjective perceptions of well-being. These results support theories such as that of Abdel-Khalek (2006), which indicated that religiosity plays an important role in the individual prediction of happiness. According to this, religiosity and its practices can predict whether an individual will have higher or lower levels of satisfaction with their own life. Regarding social support, the results of the study corroborate approaches such as that of Cohen et al. (2000), where they argue that feeling close to other people produces greater implications for both collective and individual well-being. Negative religious coping mechanisms also significantly increased the variance explained. The negative relationship between these coping mechanisms and happiness, which has been signaled recently (Vieten and Lukoff 2021), should make healthcare professionals aware of the double impact that religiousness can have on people's wellbeing.

When it comes to gender, it was found that women score higher in both religiosity and positive religious coping mechanisms compared to men. This suggests that women are slightly more religious than men and consequently develop a greater number of positive religious coping mechanisms. As mentioned by Cetrez (2011), women engage in a greater number of religious behaviors than men, being more involved in their faith. Nevertheless, in our study, being a man added 1% of explained variance of happiness. The lower score in happiness shown by women here is similar to that found in other studies (Saiz et al. 2021b). These could be explained from a cultural perspective (Wenham et al. 2020), which states that women are usually the ones who provide most of the informal care within families, so they might be suffering a bigger burden.

The present study has several limitations. First of all, the instruments used to collect quantitative data contain a limited number of items, reducing the measures taken for each of the different variables. Perhaps the use of qualitative strategies to complement this data would have been very useful. When it comes to the variable acculturation, it should be remembered that the scale has not shown the highest levels of reliability, so the interpretations should be carried out with great caution. The recruiting strategy was not random, and the final sample had biases regarding country of origin, parish, gender or age. Related to the final sample, immigrants from Venezuela are over-represented against other Latin American societies. An important amount of the participants of this study had this country as country of origin, which could have led to an under-representation of other nationalities.

As a proposal for the future, it is necessary to examine the acculturation variable with other research tools. The construct should be studied deeply to understand and facilitate contexts of acculturation, allowing a better adaptation during the migratory process. Interventions could be carried out at the state level to generate programs to support the psychological processes this group faces. Research into social support would help in seeking information on how immigrants relate to others and what sources of support are the most significant for this group. In addition, it would be relevant to study other possible elements that could play a part within this population in their perception of happiness. For example, it would also be important to include prejudices that people have about immigrants in the host country (Molero et al. 2013). This would provide a broader framework on how these people are welcomed and would determine the intervention

points to work with both immigrants and natives. In addition, it would be good if, in order to help immigrants in their adaptation to a new country, community interventions were developed that take into account the religious needs of these people, as well as the religious institutions of the host country (Kloos and Moore 2000). Finally, as Kouros and Papadakis (2018) recommended, when studying the intersection of religion and immigration, the mutual transformation of identities should be considered, and we should go beyond the dichotomy of considering religion as a bridge or a barrier.

## 5. Conclusions

The process by which a foreign person adjusts to a new country can be costly, and certainly requires numerous strategies to make it easier. In this study, we have found that for Latin American people who emigrate to Spain, religion, along with social support, is an element that should be contemplated in order to provide them with a greater perception of happiness. Religion should be considered for the design of individual and community interventions dealing with immigration.

**Author Contributions:** Conceptualization, A.M.F.-S. and J.S.; methodology, A.M.F.-S. and J.S.; software, A.M.F.-S.; validation, A.M.F.-S.; formal analysis, A.M.F.-S. and J.S.; investigation, A.M.F.-S.; resources, A.M.F.-S.; data curation, A.M.F.-S.; writing—original draft preparation, A.M.F.-S. and J.S.; writing—review and editing, A.M.F.-S., J.S., D.C. and P.J.M.; visualization, A.M.F.-S.; supervision, J.S. and P.J.M.; project administration, A.M.F.-S. and J.S. All authors have read and agreed to the published version of the manuscript.

**Funding:** This research received no external funding.

**Institutional Review Board Statement:** Not available.

**Informed Consent Statement:** Informed consent was obtained from all subjects involved in the study.

**Data Availability Statement:** The data presented in this study are available on request from the corresponding author.

**Conflicts of Interest:** The authors declare no conflict of interest.

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
