# Peer review of "The Impact of Religion and Social Support on Self-Reported Happiness in Latin American Immigrants in Spain"

_religions, doi:10.3390/rel13020122_

Round 1

Reviewer 1 Report

The article has merit in the sense that it presents original data from Region of Madrid, Spain, to support that religion is an important aspect of immigrant integration in a host society. The article presents original data to propose that religion is a factor that plays an important role in the integration of immigrants.

The literature presented in the introduction is sufficient and relevant. More relevant literature would benefit the article. Considering that the article is concerned with the relationship(s) between immigration, integration and religion, could benefit more the overall well-written article. While there is ample literature about this issue, I would suggest Kouros and Papadakis’ (2018) work on the religious integration of various immigrant communities in Cyprus (a very similar topic), who also give an overview of this theoretical discussion. Moreover, the authors will find beneficial to their arguments much of van Tubergen’s and Connor’s work on the topic (indicatively van Tubergen & Sindradóttir 2011, and Simsek, van Tubergen & Fleischmann 2021, Connor 2009, Connor and Koenig 2013).

In terms of methodology and results, the article is well-written and supported. Every aspect of the methodology and tools utilized is clearly explained.

Author Response

Thank you so much for the literature recommendations. We have introduced these new references which indeed improve our theoretical background and discussion.  We reviewed all the given references, and specifically we used Kouros and Papadakis (2018); Connor and Koenig (2013) and Simsek, van Tubergen and Fleischmann (2020). It was very interesting to read, so thank you very much for the recommendation.

We really appreciate these kind remarks on our manuscript, which indeed inspires us to improve our work.

Reviewer 2 Report

The paper shows evidence of a significant relationship between religion, social support and the perception of happiness among the Latin American immigrant population in Madrid (Spain) that responded to the survey administered. Therefore, it is reasonable to think that religion together with social support will help to overcome the difficulties involved in the immigration process and integration in a different cultural context. In a society where migration flows are increasing, taking this into account can have important practical consequences. In this sense, the paper makes an interesting contribution.

The paper has two weaknesses that should be taken into account and, to some extent, remedied.

1) Definition of the concepts studied.

1.1) Definition of “religion” and “religiosity”. The paper rightly points out the ambiguity of the term "religion" and the need to opt for a definition; the author favours Koenig's definition (107-110). However, the paper generally does not speak of “religion” but of “religiosity” (196, 276 and so on); is “religion” the same as “religiosity”? There is a very brief allusion to the description of religiosity in line 284: “intrinsic religiosity (...) refers to the degree of individual commitment to religion". This comment would lead one to think that the author differentiates between religion and religiosity but it is very vague; on the contrary, it seems to implicitly assume that the two terms are equivalent. The current literature is far from identifying the two terms; “religion” is often associated with the more institutional level (public dimension) and “religiosity” with personal experience (private dimension), although there are other variants in the meanings; it also has crossovers of meaning with the terms "spiritual", "spirituality". One can legitimately choose to identify religion and religiosity, but this requires a brief argumentation and terminological clarification.

1.2)  Definition of "coping mechanism of religious". This concept is not clearly defined. A descriptive approximation can be what is said in lines 54-55. Without further specification, "coping mechanism of religious" appears rather as a function of religion (or “religiosity”). In fact, the examples in lines 266 and 268 are clearly religious content expressed in terms of personal experience or desire and fall within the definition of religion that the author has chosen (Koenig's definition). In this sense, it is logical that there is such a high correlation between religiosity and "positive religious coping" and no relationship with "negative religious coping" (Table 4). It seems that the term "religion" is being linked to one of the "functions of religion". If one insists on keeping the concept "coping mechanism of religious" as independent of the term religion (or religiosity) some kind of terminological clarification is needed to clarify the specificity.

2) The sample.

The author gives an account of how the sample was constructed and how the individuals in the sample were contacted. The sample has not been constructed as a representative sample of the immigrant population in Spain, not even in Madrid.  The selection has been made according to the criteria indicated but from the population that responded to the survey; this population is not necessarily representative of the Latin American immigrant population in Madrid. This introduces some important biases that must be taken into account:

  1. a) Latin America is not homogeneous and the area of origin may have a bearing on the response. Venezuela is over-represented and other Latin American societies are under-represented.
  2. b) Access was through parishes in Madrid, which implies a certain type of population. A significant proportion of Latin American immigrants are probably linked to the Protestant evangelical tradition and especially to the Pentecostal churches; Catholic parishes do not seem to be a good starting point for accessing this type of non-Catholic population.

Logically, this point cannot be changed in the study, but it would be convenient to indicate it in the limitations of the study. It does not invalidate the general statement, but it implies limits to the generalisations that can be made.

Minor issues

  1. Within section 1 (Introduction) there are the following sub-sections: 1.1., 2.1., 3.1. Moreover, there seems to be a missing section between the introduction and section 2 (Method); it does not seem logical for the introduction to be so long and to have sub-sections. This part seems to be poorly structured.
  2. Perhaps it would be useful to add a short final section containing the conclusions of the work carried out; these are partly mentioned in the "Discussion" section, but the final conclusions are not very systematised.
  3. On lines 11-12 the following statement appears: "It was observed that religiosity, social support, negative religious coping mechanisms and gender were related to the perception of happiness". Based on the central statement of the article and, specifically, on the considerations made in lines 393, 404-405, 417, it seems that the statement should change "negative religious coping mechanisms" to "positive religious mechanisms". Perhaps the wording in the abstract refers to the explanation of variance as it appears in lines 366 and 367; if that is the idea that is intended to be expressed in the abstract, the wording does not seem clear.
  4. In lines 174-176 the following statement appears: "in countries like Estonia, the results show an inverse relationship between religion and happiness, that is, that practicing a religion is not relevant when determining a person's well-being". This is not an inverse relationship (the more religion the less happiness), but a lack of relationship.
  5. The sample was taken in the Autonomous Community of Madrid. The title talks about Spain, which leads one to think that it is a sample of Latin Americans in Spain that has been surveyed. This information is already included in lines 216-217 but it seems to be a bit marginal and may go unnoticed. Perhaps it would be important to make a brief comment in the discussion of the data, since this is one of the limitations of the research carried out, especially with a view to any kind of generalisation.
  6. Reliability of the applied Spanish measurement surveys. In the case of "Coping mechanisms of religious origin", "Acculturation" and "Subjective happiness" it is indicated that the surveys have been validated in the Spanish version; this is not the case for the rest of the variables measured. If the Spanish version has not been validated, a brief indication of the precautions taken should be given.

Author Response

We really appreciate these kind remarks.

Thanks to this comment we have introduced reviewer´s clarification along with Koenig´s definition. We hope it serves better to the study and to the readers.

In order to improve the beggining of our article, we have introduced Wortmann´s definition of “religious coping”. We have restructured the introduction, so it becomes better for its own understanding.

Thank you very much for this. Indeed, we couldn’t do anything afterwards but we pointed it out as a limitation of our study so the readers can have it in mind while making their own conclusions.

Thank you so much for this suggestion that allows us to improve our Discussion and to concentrate our results in a new section.

The reviewer is right; the idea that was intended to be expressed in the abstract was the explanation of variance, but we have eliminated the word “negative”, so it doesn’t cause any confusions. Thanks for pointing it out.

We added some lines explaining that the participants were actually taken from different parishes, not only in Madrid but in the rest of Spain. It’s true that most of the participants were from the region of Madrid because we had more access to them.

All of the instruments used along the study were validated in Spanish versions before. We cleared it out better so readers do not get confused. Thank you very much.

Thank you for all the kind comments.
